# A Metabonomic View on Wilms Tumor by High-Resolution Magic-Angle Spinning Nuclear Magnetic Resonance Spectroscopy

**DOI:** 10.3390/diagnostics12010157

**Published:** 2022-01-10

**Authors:** Ljubica Tasic, Nataša Avramović, Melissa Quintero, Danijela Stanisic, Lucas G. Martins, Tassia Brena Barroso Carneiro da Costa, Milka Jadranin, Maria Theresa de Souza Accioly, Paulo Faria, Beatriz de Camargo, Bruna M. de Sá Pereira, Mariana Maschietto

**Affiliations:** 1Laboratory of Chemical Biology, Institute of Chemistry, University of Campinas (UNICAMP), Campinas, Sao Paulo 13083-970, Brazil; meliquies@gmail.com (M.Q.); dacici.stanisic@gmail.com (D.S.); lgmartins1984@gmail.com (L.G.M.); tassiabrena@gmail.com (T.B.B.C.d.C.); 2Faculty of Medicine, Institute of Medical Chemistry, University of Belgrade, Višegradska 26, 11000 Belgrade, Serbia; natasa.avramovic@med.bg.ac.rs; 3Institute of Chemistry, Technology and Metallurgy, Department of Chemistry, University of Belgrade, Njegoševa 12, 11000 Belgrade, Serbia; milkaj@chem.bg.ac.rs; 4National Bank of Tumors of National Cancer Institute (BNT-INCA), Rio de Janeiro 20231-091, Brazil; maccioly@inca.gov.br; 5Department of Pathology, Federal University of Rio de Janeiro (UFRJ), Rio de Janeiro 21941-901, Brazil; pauloafaria@gmail.com; 6Clinical Research Department, National Cancer Institute (INCA), Rio de Janeiro 20231-091, Brazil; bdecamar@terra.com.br (B.d.C.); pereira.brunams@gmail.com (B.M.d.S.P.); 7National Laboratory of Biosciences (LNBio), National Center for Research in Energy and Materials (CNPEM), Campinas, Sao Paulo 13083-100, Brazil; marianamasc@gmail.com

**Keywords:** Wilms tumor (WT), NMR-metabonomics, metabolic pathways

## Abstract

Pediatric cancer NMR-metabonomics might be a powerful tool to discover modified biochemical pathways in tumor development, improve cancer diagnosis, and, consequently, treatment. Wilms tumor (WT) is the most common kidney tumor in young children whose genetic and epigenetic abnormalities lead to cell metabolism alterations, but, so far, investigation of metabolic pathways in WT is scarce. We aimed to explore the high-resolution magic-angle spinning nuclear magnetic resonance (HR-MAS NMR) metabonomics of WT and normal kidney (NK) samples. For this study, 14 WT and 7 NK tissue samples were obtained from the same patients and analyzed. One-dimensional and two-dimensional HR-MAS NMR spectra were processed, and the one-dimensional NMR data were analyzed using chemometrics. Chemometrics enabled us to elucidate the most significant differences between the tumor and normal tissues and to discover intrinsic metabolite alterations in WT. The metabolic differences in WT tissues were revealed by a validated PLS-DA applied on HR-MAS T_2_-edited ^1^H-NMR and were assigned to 16 metabolites, such as lipids, glucose, and branched-chain amino acids (BCAAs), among others. The WT compared to NK samples showed 13 metabolites with increased concentrations and 3 metabolites with decreased concentrations. The relative BCAA concentrations were decreased in the WT while lipids, lactate, and glutamine/glutamate showed increased levels. Sixteen tissue metabolites distinguish the analyzed WT samples and point to altered glycolysis, glutaminolysis, TCA cycle, and lipid and BCAA metabolism in WT. Significant variation in the concentrations of metabolites, such as glutamine/glutamate, lipids, lactate, and BCAAs, was observed in WT and opened up a perspective for their further study and clinical validation.

## 1. Introduction

Metabolomics depicts and fingerprints the final biochemical effects on molecules with molar masses lower than 2 kDa [1,2,3,4,5,6,7] by comparing groups, such as healthy and diseased. One of the greatest advantages of high-resolution magic-angle spinning (HR-MAS) nuclear magnetic resonance (NMR) spectroscopy is the possibility of analyzing ex vivo samples that mimic on-site analysis and, thus, providing clues on changes in concentration and type of metabolites in the given moment. Therefore, metabonomic studies in pediatric cancers might be a powerful tool to investigate specific metabolic alterations in biochemical pathways and may serve as a potential tool to improve cancer diagnosis [1,2]. Wilms tumor (WT) is the most common renal cancer in childhood, corresponding to about 5% of all cases of pediatric cancers diagnosed between 0 and 14 years of age [8,9]. A remarkable characteristic of WT is the relatively low frequency of genetic alterations, with up to 35 genes mutated in approximately 30% of the cases. The most recurrent mutated gene does not exceed the 14% rate [10]. Epigenetic alterations are identified in up to 70% of WT cases, including alterations in the miRNA processing machinery and DNA methylation [10,11]. Although recent studies indicate that genetic and epigenetic abnormalities promote cell metabolism alterations in WT, so far, little is known about the role of specific metabolic pathways in WT development. Feichtinger et al. [12] showed that the histological subtypes of WT have different energy metabolism. Blastemal and epithelial components presents a normal mitochondrial mass, while the stromal subtype reveals a loss of mitochondrial content compared to normal kidney tissue, indicating a deficiency of all oxidative phosphorylation (OXPHOS) enzymes, low porin expression, and citrate synthase activity [12]. Therefore, alterations in glycolytic and lipid metabolism pathways are expected in WT, with a very high conversion of glucose to ATP and lower rates of catabolism of triacylglycerols (TAGs), and fatty acids (FAs) but higher anabolism of phospholipids (PL) [6]. Kidney metabolic alterations were identified as a trigger for cancer in adults, as reported by Linehan et al. [7], who went even further by identifying kidney cancer as a direct metabolic disease. On the other side, NMR-metabonomics in WT are scarce. For example, there are some data on liquid NMR urine metabolomics where samples of WT showed an increase in the concentrations of the nonpolar amino acids alanine, leucine, and isoleucine (Ala, Leu, Ile), and also iso-valerate, 2-hydroxybutyrate, 2-oxoisovalerate, glucose, dimethylamine, and 2-oxoglutarate and a decrease in the concentrations of creatine, creatinine, acetate, and citrate when compared to controls [13]. As far as we are aware, there is no data on intact WT tissues using NMR. Consequently, the aim of our study was to explore the HR-MAS ^1^H-NMR metabonomics of WT and normal kidney (cortex—NK) samples taken from the same patients to identify the metabolic pathways and specific metabolites that are altered in WT samples and are potentially related to Wilms tumorigenesis. HR-MAS-based metabonomics, through the identification of WT metabolite hallmarks might provide insights into patterns of responsiveness to treatment and relapse and lead to personalized treatment. Finally, tissue biopsy by NMR may allow us to understand the molecular changes occurring in the tumor in real-time, derived from intratumor heterogeneity and/or therapeutic pressure and may greatly add to the oncological research.

## 2. Materials and Methods

### 2.1. Samples

This study included frozen tissues from eleven WT patients diagnosed and treated with pre-chemotherapy and three treated with upfront surgery. The samples were collected at the time of surgery and stored frozen at −80 °C until NMR analysis at the National Bank of Tumors of INCA (BNT-INCA, Rio de Janeiro, RJ, Brazil). All samples were reviewed by a single pathologist who confirmed and selected fourteen WT and seven NK tissue samples for experimental analyses. The WT tissue samples were taken from patients diagnosed as cancer clinical stage III (4/14 cases), when the cancer has spread to lymph nodes in the abdomen or pelvis but not to more distant lymph nodes, or as clinical stage II (10/14 cases). The NK samples matched WT samples diagnosed as clinical stage III (2/7 cases) or II (5/7 cases). The WT samples were taken from equal populations of female and male patients (50%), while the NK samples were taken mostly from female patients (5/7 cases). The Research Ethics Committee from INCA approved this study (CAAE 09981018.3.0000.5274) with informed consent from parents or the children’s legal guardians.

### 2.2. Metabonomics by Nuclear Magnetic Resonance Spectroscopy

Fourteen WT and seven NK tissue samples were analyzed by high-resolution magic-angle spinning nuclear magnetic resonance (HR-MAS NMR) spectroscopy. ^1^H-NMR spectra were performed using a Bruker Avance spectrometer (Bruker BioSpin, Germany) operating at 400 MHz and equipped with the double nuclei 4 mm probe for HR-MAS. One-dimensional water-suppressed ^1^H-NMR spectra were recorded with the nuclear Overhauser effect spectroscopy (NOESY1D) pulse sequence and 256 repeats, and the T_2_-edited spectra were recorded using the CPMG (Carr–Purcell–Melboom–Gill) pulse sequence with 128 repetitions. All spectra were performed at a magic-angle spinning frequency of 4 kHz and 298 K while the chemometrics analysis was executed through the open-access platform MetaboAnalyst (www.metaboanalyst.ca, accessed on 13 November 2021). Information about processing NMR spectra, pre-processing chemometrics data, and metabolites’ assignments were previously depicted [14,15,16].

### 2.3. Statistical Analysis of NMR Data

The matrices were constructed with 21 spectra (14 for the WT group and 7 for the NK group) and 2800 variables for the parts of the aliphatic region of the T_2_-edited ^1^H-NMR spectra, δ 0.50–4.50, where, according to the subtraction of the mean spectra of WT and NK, most of the differences were observed. Two principal component analyses (PCA) were performed, one using all spectra, to explore the inherent groupings within samples and to identify outliers, and the other with the spectra of the paired samples of WT and NK tissues from the same patients (14 spectra in total, 7 for WT tissues and 7 for NK tissues). All WT data were modeled with the supervised method of partial least squares discriminant analysis (PLS-DA) to find the metabolite differences between the groups. According to the PLS-DA models, variable importance in projection (VIP) scores were estimated to depict the most different chemical shifts among the analyzed samples. Finally, an orthogonal projections to latent structures discriminant analysis (oPLS-DA) was employed.

## 3. Results

We evaluated 14 WT and 7 NK tissue samples taken after surgery (3 WT samples) and preoperative chemotherapy (11 WT and 7 NK) by HR-MAS ^1^H-NMR to characterize the intact tissue metabolites and metabolic fingerprints of Wilms tumor. WT is the most frequent kidney cancer. It has a prevalence rate of 8.33 per million children in Brazil [17,18,19,20,21]. Therefore, the investigated cohort of samples is small but significant in terms of the rarity of the disease. It is especially important to emphasize that, at the time of sampling, paired WT and NK underwent the same treatment with pre-chemotherapy.

Two types of 1D NMR experiments by HR-MAS were performed to snapshot tissues. Nuclear Overhauser enhancement spectroscopy (NOESY1D) was preformed to measure all hydrogen species in the samples, which were then edited with a T_2_-filter (CPMG), where low-molecular-mass metabolites were studied and the signals from macromolecules were attenuated.

An inspection of the data revealed the most significant spectral differences among tissue samples (WT and NK), revealing that T_2_-edited aliphatic and aromatic regions were the ones with the greatest metabolic variations in cancer tissue (Figure 1). Further, the observed differences were confirmed by the chemometrics analysis results (Figure 2 and Figure 3 and Appendix A).

The WT samples were similar metabolically among different patients and showed a similar heterogeneity to NK samples, as could be seen in chemometrics results shown in Figure 2 and Figure 3 and Appendix A.

The partial least squares discriminant analysis (PLS-DA) results are shown in Figure 2 and Figure 3, for 21 and 14 analyzed samples, respectively. It is worth saying that the cluster analysis presented in Figure 3B and in the heatmap (Figure 4B) also show the separation of groups without the misclassification of the samples WT or NK. Furthermore, WT samples appeared to be more similar among each other than to NK, independent of the fact that the NK and WT tissues were taken from the same individual.

The metabolites that distinguished WT from NK samples were revealed by validated PLS-DA (Accuracy: 0.877, R^2^: 0.690, and Q^2^: 0.577), and among them, sixteen metabolites (Figure 1 and Figure 2B) showed the highest contributions to group separation, with variable importance in projection (VIP) values higher than 2.5. Using the 1D and 2D NMR data (TOCSY), those were assigned to lipids, glucose, aliphatic branched-chain amino acids (BCAAs), and other metabolites (Figure 1 and Figure 2B). It is worth stating that the oPLS-DA model for the same datasets is illustrated in the Appendix A.

There are differences in metabolites’ concentrations in WT compared to the NK tissue samples (Figure 2B and Appendix A), from which 13 metabolites showed increased and 3 decreased concentrations in the WT samples. The concentrations of the BCAAs, including valine, leucine, and isoleucine were decreased in WT tissue samples compared to controls. However, WT samples showed higher levels of lipids, lactate, and glutamine/glutamate versus controls, as illustrated in Figure 2B and Figure 4.

## 4. Discussion

The metabolite concentrations that differ in WT and NK tissue samples may point to altered metabolic pathways and may be potential diagnostic candidates. WT tissues showed altered metabolic pathways that involve the TCA cycle, lipids, glycolysis, glutaminolysis, and branched-chain amino acid (BCAA) metabolism when compared to NK samples [22,23].

Cancer cells have an enhanced requirement for glucose as a nutrient, and instead of converting it through pyruvate to acetyl-CoA via the TCA cycle of energy production, glucose is mainly converted to lactate by glycolysis [23,24]. A higher level of lactate was identified in the investigated WT tissue samples, demonstrating altered glycolysis. Tumor cells present a highly activated glycolytic pathway in the presence of oxygen (Warburg effect) that contributes to cell proliferation [24]. Tumor cells consume glutamine at unusually high rates, and its metabolism, called glutaminolysis [24], is involved in the neoplastic transformation, since its inhibition decreases cell proliferation [25]. WT samples presented an increased concentration of glutamine and glutamate supporting enhanced glutaminolysis. The conversion of glutamine to glutamate by glutaminase (GLS1 or GLS2), which is further transformed to α-ketoglutarate (α-KG) by glutamate dehydrogenase (GLUD1 or GLUD2) or aminotransferases, promotes the TCA metabolism that contributes to sustaining the energy used for cancer cell proliferation [25,26,27]. The TCA cycle also promotes lipid synthesis through the conversion of α-KG in citrate by reductive carboxylation. The increased concentrations of lipids in WT tissue samples demonstrated increased lipid demand in cancer cells due to the requirement for the synthesis of cell membranes whose components are phospholipids (PL) (Figure 5) [28].

Main cell membrane PL is sphingomyelin including choline in its content. Recently MacLellan et al. reported decreased concentrations of urinary creatine and creatinine in patients with WT [13]. Creatinine is a choline precursor, supporting the fact that with increasing lipid demand in cancer tissues, urinary lipid concentration decreases, leading to a reduction in urinary choline and, consequently, in creatine and creatinine in WT patients, similar to other cancers [24,28].

In contrast to the increase in concentrations of lactate, glutamine/glutamate, and lipids, concentrations of three BCAAs decreased in WT samples compared to controls. Indeed, BCAAs are essential amino acids taken up through the diet due to the impossibility of synthesis by human cells [29]. Glutamate is a co-product in the first step of BCAA catabolism with the conversion of BCAAs to branched-chain keto acids (BCKA) by branched-chain aminotransferase (BCAT1 in cytosol or BCAT2 in mitochondria). BCAA-originated glutamate is a source of the nitrogen used for the synthesis of macromolecules such as nucleotides and the proteins necessary for cancer cell growth (Figure 5) [24,30,31,32]. Altered BCAA metabolism has an important role in cancer progression and the crucial enzymes in the BCAA metabolic pathway might be possible prognostic and diagnostic biomarkers in human cancers. Our study also supports the finding that BCAAs in WT tissue can play a role as a diagnostic cancer biomarker. On the other hand, MacLellan et al. showed increased concentrations of urinary BCAAs in WT patients [13]. The contrasting results may be a consequence of the different treatments the patients were subjected to. Most samples from our study came from patients that underwent chemotherapy. Enhanced intracellular α-KG levels in WT by promoted glutaminolysis also increases the activity of another α-KG-dependent dioxygenase, such as the ten-eleven translocation 2 (TET2) DNA demethylase and leads to DNA hypomethylation [23,33]. This finding is in agreement with the recently reported data by Guerra et al. which demonstrated a hypomethylation profile in WT compared to NK through global gene expression and methylation analyses [17]. These findings suggest that metabolic pathways in WT are altered by diverse oncogenes and tumor suppressor genes such as MYC, HIF, TGFβ, WNT, and mTOR signaling pathways promoting the upregulation of many metabolic enzymes and transporters (Figure 5). In this way, transporters of glucose (GLUT) and glutamine (ASCT2) and metabolic enzymes, including glutaminase (GLS1 and GLS2), glutamate dehydrogenase (GLUD1 or GLUD2), and branched-chain aminotransferase (BCAT1 and BCAT2), might have an important role in the diagnosis and treatment of WT and could be attractive targets for further exploration and research.

A limitation of this study is that the samples used had received preoperative chemotherapy, according to the SIOP protocol [34,35,36]. Thus, we cannot exclude the possibility that at least part of the changes in metabolites levels was induced by the treatment. Moreover, the number of cases prevents us from comparing WT subgroups, such as histological subtypes and clinical stages. Another limitation is that we cannot directly compare our findings with previous studies that evaluated metabolites in patients with WT as the sources used are different (tissue vs. urine), preventing some validation of our findings.

## 5. Conclusions

In our study, we summarize that altered glycolysis, glutaminolysis, TCA cycle, lipid metabolism, and BCAA metabolism in WT tissue by alterations of the concentrations of several metabolites such as glutamine/glutamate, lipids, lactate, and BCAAs, which might be considered as potential diagnostic WT biomarkers. Metabolite levels in WT tissue are changed due to the requirement of cancer cells for nutrients and the building blocks to survive and proliferate, activating diverse signaling pathways and oncogenes, which modify metabolic pathways through the upregulation of metabolic enzymes. Considering the important role of glycolysis, glutaminolysis, and BCAA metabolic reprogramming in WT, the metabolites, and their related metabolic pathways might be involved with Wilms tumorigenesis and deserve to be explored in prospective studies. The identified differences among the metabolites in WT tissues may lead, not just to the identification of WT biomarkers with the potential for early cancer diagnostic and prognostic purposes, but also to a tailor-made treatment.

## Figures and Tables

**Figure 1 diagnostics-12-00157-f001:**
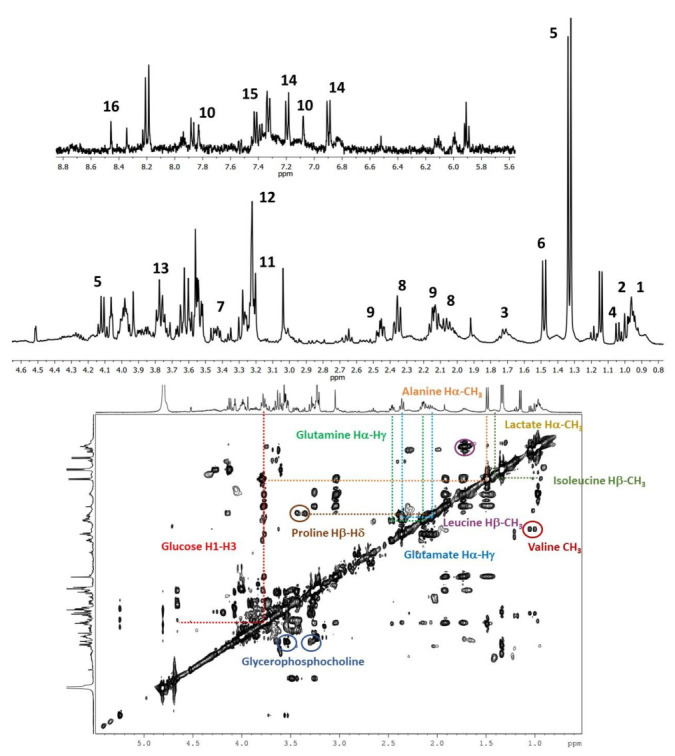
^1^H-NMR HR-MAS spectrum of WT tissue sample acquired using a CPMG (*cpmgpr1d*) pulse sequence; two regions are presented in the lower panel at 0.50 and 4.70 ppm, and, in the upper panel, the region between 5.0 and 9.00 ppm is amplified 20 times. Sixteen metabolites were identified: fatty acids (CH_3_-) (1), isoleucine (2), leucine (3), valine (4), lactate (5), alanine (6), proline (7), glutamate (8), glutamine (9), histidine (10), choline (11), glycerophosphocholine (12), glucose (13), tyrosine (14), phenylalanine (15), and formate (16). At the bottom the ^1^H-^1^H TOCSY spectrum is shown (aliphatic region, 0.00–5.50 ppm) with marked correlations.

**Figure 2 diagnostics-12-00157-f002:**
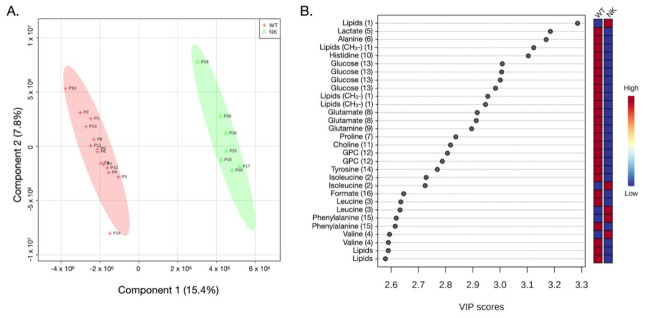
(**A**) PLS-DA score plot in 2D of the HR-MAS ^1^H-NMR CPMG data, showing 21 tissue samples, 14 WT and 7 NK. The WT tissue samples are shown with the red crosses and the normal kidney (NK) tissue samples are shown with green triangles. Accuracy: 0.877, R^2^: 0.690, and Q^2^: 0.577. (**B**). Variable importance in projection (VIP) scores greater than 2.5 show the important metabolites (1–16 as illustrated in Figure 1) that are discriminatory for WT vs. NK in the PLS-DA model.

**Figure 3 diagnostics-12-00157-f003:**
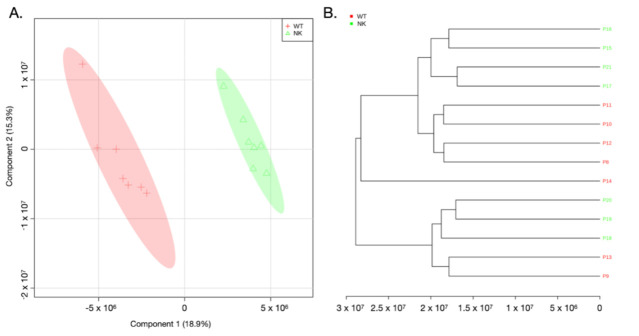
(**A**) PLS-DA score plot in 2D of the HR-MAS ^1^H-NMR CPMG data, showing 14 paired tissue samples, 7 WT and 7 NK. The WT tissue samples are shown with red crosses and the normal kidney (NK) tissue samples are shown with green triangles. Accuracy: 0.774, R^2^: 0.902, and Q^2^: 0.430. The VIP scores greater than 2.7 corresponded to metabolites from the aliphatic region (0.50 to 4.50 ppm, metabolites 1–9 and 11–13, see Figure 1 and Figure 4). (**B**) Hierarchical Cluster Analysis (HCA) shows a graphical representation of the PCA scores for component 1 (describing 17.1% of the total variance) plotted against component 2 (13% of the total variance) and shows the similarities or differences among the tissue samples, wherein the samples that form clusters show similarities, and samples that are found at greater distances are dissimilar. The WT P14 is a unique sample with incidence of metastasis. NK samples P18 and P19 were from the patients with WT clinical stage III. Similarities were observed among the WT samples and among the WT and NK samples from the same individuals.

**Figure 4 diagnostics-12-00157-f004:**
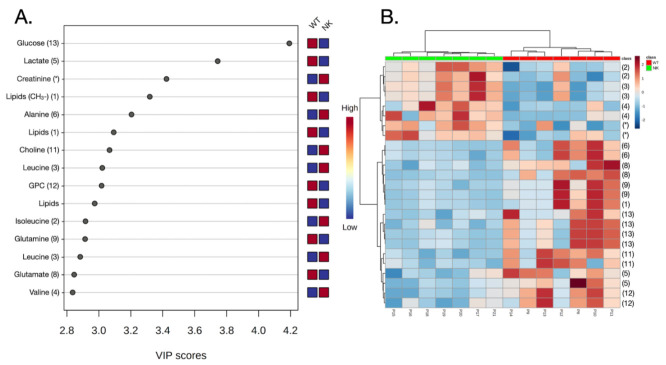
(**A**) Variable Importance in Projection (VIP) scores greater than 2.7 obtained in the ^1^H-NMR CMPG HRMAS model by PLS-DA. The most significant concentration variations were observed for lipids, lactate, and glucose, as could be seen for the model when the seven WT and seven NK paired samples were analyzed. The numbers given in brackets correspond to the metabolites’ assignments shown in Figure 1. (**B**) Cluster analysis presented as a heatmap. The WT samples are shown in red, and the numbers in brackets correspond to the metabolites’ assignments shown in Figure 1.

**Figure 5 diagnostics-12-00157-f005:**
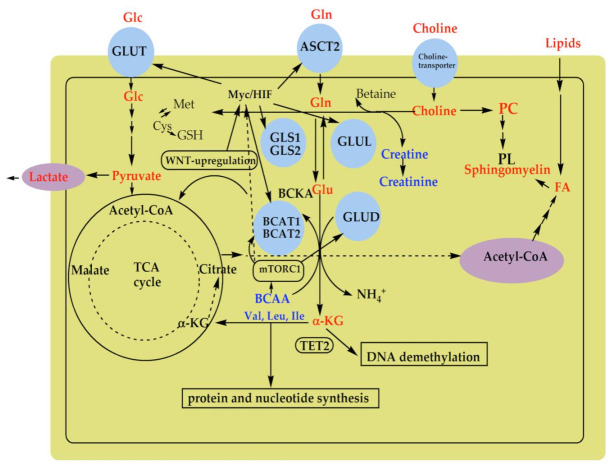
Illustration of the main metabolic pathways that are reported to be altered in WT. Metabolites with altered concentrations in WT tissues included glucose (Glc), glutamine (Gln), glutamate (Glu), α-ketoglutarate (α-KG), phosphatidylcholine (PC), phospholipids (PL), branched-chain amino acids (BCAAs, such as valine (Val), leucine (Leu), isoleucine (Ile)), as well as, cysteine (Cys), methionine (Met), and glutathione (GSH). Metabolic pathways in WT are altered by oncogenes and tumor suppressor genes such as *MYC*, hypoxia-inducible factors (*HIF)*, transforming growth factor-beta (TGFβ)-WNT, and complex 1 of the mammalian target of rapamycin (mTORC1) signaling pathways promoting upregulation of many metabolic enzymes including glutaminase (GLS1 and GLS2), glutamine synthetase (GLUL), glutamate dehydrogenase (GLUD), branched-chain aminotransferase (BCAT1 and BCAT2), ten-eleven translocation 2 DNA demethylase (TET2), and transporters such as transporter of glucose (GLUT) and glutamine (ASCT2).

## Data Availability

The authors confirm that the data supporting the findings are included in the article and the set of raw data that support the reported findings is available from the corresponding author upon request.

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
