# Peer review of "A Metabonomic View on Wilms Tumor by High-Resolution Magic-Angle Spinning Nuclear Magnetic Resonance Spectroscopy"

_diagnostics, 2022, doi:10.3390/diagnostics12010157_

Round 1

Reviewer 1 Report

The paper deals with a  pediatric cancer NMR-metabonomics study focused on a kidney tumor  (Wilms tumor, WT). An interesting use of  high-resolution magic angle spinning nuclear magnetic resonance (HR-MAS NMR) based metabonomics of WT and normal kidney (NK) samples is proposed with the clear advantage of direct metabolic profiles observation in the tissues. A good discrimination between the metabolic profiles of tumor and normal tissues was obtained and studied by chemometrics with an expert use of supervised methods. The observed differences in metabolites were correctly discussed in term of possibly involved biochemical pathways. The paper is well focused and written and deserves publication in the journal.  Minor points that could be addressed in order to further improve an already sound work are:

1) possible identification (despite the limited number of samples) of the leading factors for the orthogonal dispersion observed in figure 1 (sex, age, disease stage…?);

2) brief discussion on the possibility to correlate this study, which appears essentially related to tumorigenesis connected investigations, also to potential diagnostic applications.

Author Response

Responses to #1 Reviewer comments 

The paper deals with a pediatric cancer NMR-metabonomics study focused on a kidney tumor (Wilms tumor, WT). An interesting use of high-resolution magic angle spinning nuclear magnetic resonance (HR-MAS NMR) based metabonomics of WT and normal kidney (NK) samples is proposed with the clear advantage of direct metabolic profiles observation in the tissues. A good discrimination between the metabolic profiles of tumor and normal tissues was obtained and studied by chemometrics with an expert use of supervised methods. The observed differences in metabolites were correctly discussed in term of possibly involved biochemical pathways. The paper is well focused and written and deserves publication in the journal. 

Answer: Thank you for the comments and thorough analysis of our manuscript.

Minor points that could be addressed in order to further improve an already sound work are: 

  1. possible identification (despite the limited number of samples) of the leading factors for the orthogonal dispersion observed in figure 1 (sex, age, disease stage...?); 

Answer: Thank you for the observation. We did not observe sex or age influences on the dispersion of samples in orthogonal T score (y-axis). There was some dispersion because of the disease stage (II or III, and IV/distant metastasis), but as you have pointed out, there is a limited number of cases to explore such differences in T scores. It could be noted that 2 NK samples and 4 WT samples showed differences in comparison with other samples, clinical stage I which could be related to the clinical stage. 

2) brief discussion on the possibility to correlate this study, which appears essentially related to tumorigenesis connected investigations, also to potential diagnostic applications. 

Answer: Thank you for the observation. We have added short text regarding the potential application in diagnostics. The gained insights can be explored further in the assessment of treatment response and relapse.

Reviewer 2 Report

In this manuscript Tasic et al. report a metabolomics study of Wilms tumor using solid state NMR spectroscopy. The manuscript is well-written, and could provide an important contribution for the understanding of metabolic rewiring in Wilms tumor given that revisions are implemented. Nevertheless, there are substantial revisions necessary to substantiate some of the conclusions, discussion of the results, and the motivation for the study.

  • Motivation: the authors should clarify what the added value of their study is. To me it is unclear how the metabolomics results should be further utilized in the future. Potential questions could be - are there any patient-to-patient heterogeneities due to different tumor subtypes which could be studied? Could the metabolites used to assess relapse or treatment response?
  • O-PLS-DA: reporting solely the plots shown in Fig. 1 is insufficient to judge the quality of the model. Proper cross-validation (Q2, p-values) should be reported and a representation in which the covariance and R2s are plotted as function of chemical shifts.
  • Spectrum in Figure 2: the authors should carefully check the metabolite assignments shown in Figure 2. Some of the metabolites seem to be mislabeled. For example, looking at the multiplet, glutamine (#9) seems to me more like glutamate (according to my own experiences Gln concentrations are lower in kidney compared to Glu). Glucose is better identified via the H1’, but this is invisible in the spectrum because the authors truncated the spectrum at 5.6 ppm. Showing a reduced spectrum (covariance/R2) will help to better identify metabolites. The discussion needs to be adapted accordingly.
  • Figure 3 and 4: showing volcano plots and heatmaps using features is meaningless without metabolite ID. The authors should integrate well-isolated metabolite signals and use these integrals for further analysis or determine concentrations/integrals using line shape fitting with programs such as Chenomx or other comparable software.

Author Response

Responses to #2 Reviewer comments 

In this manuscript Tasic et al. report a metabolomics study of Wilms tumor using solid state NMR spectroscopy. The manuscript is well-written, and could provide an important contribution for the understanding of metabolic rewiring in Wilms tumor given that revisions are implemented. Nevertheless, there are substantial revisions necessary to substantiate some of the conclusions, discussion of the results, and the motivation for the study. 

Answer: Thank you for the comments and thorough analysis of our manuscript. We have added new information and explained the points raised by the reviewers. We hope that the revised version of the manuscript is up to expectations. 

Motivation: the authors should clarify what the added value of their study is. To me it is unclear how the metabolomics results should be further utilized in the future. Potential questions could be - are there any patient-to-patient heterogeneities due to different tumor subtypes which could be studied? Could the metabolites used to assess relapse or treatment response? 

Answer: Thank you for pointing to these important features. The motivation to perform this study was the lack of information about metabolomics in WT tissue, and to verify if the cancer alterations can be observed using HRMAS NMR on intact samples. The gained insights can be explored further in the assessment of treatment response and relapse. We have modified the last paragraph of the introduction section and explained better aims and motivation. 

O-PLS-DA: reporting solely the plots shown in Fig. 1 is insufficient to judge the quality of the model. Proper cross-validation (Q2, p-values) should be reported and a representation in which the covariance and R2s are plotted as function of chemical shifts. 

Answer: We agree with your opinion. Thank you for the observations. We made the mistake of not proving the data in the previous version of the manuscript. We have added the new and important information in the text. Also, oPLS-DA graphs were substituted with the PLS-DA in the main text, while oPLS-DA were placed in Supplementary Information. All variables that were identified as important for distinguishing the WT from the NK group were validated. Those that showed high magnitude and high reliability - in p[1] vs. p(corr)[1] - according to loadings S-plot (showing the variable importance in a model, combining the covariance and the correlation (p(corr)) loading profile), were discussed. 

Spectrum in Figure 2: the authors should carefully check the metabolite assignments shown in Figure 2. Some of the metabolites seem to be mislabeled. For example, looking at the multiplet, glutamine (#9) seems to me more like glutamate (according to my own experiences Gln concentrations are lower in kidney compared to Glu). Glucose is better identified via the H1’, but this is invisible in the spectrum because the authors truncated the spectrum at 5.6 ppm. Showing a reduced spectrum (covariance/R2) will help to better identify metabolites. The discussion needs to be adapted accordingly. 

Answer: As suggested, we have checked the assignments, and added a new part to Figure 1 in the revised version of the manuscript, a correlation spectrum (1H-1H TOCSY), which has been used to confirm the assignment of metabolites. The assignments of the metabolites, glutamine, and glutamate signals were correctly performed. We agree that the anomeric H signal of glucose is not overlapped with any other signal and is the most used for glucose identification. However, the NMR CPMG pulse sequence used for our studies promotes the elimination of the water signal through pre-saturation and, often, the glucose anomeric signal is also eliminated.

Glutamate

2.04 m; 2.119 m; 2.341 m; 3.748 dd (J=7.186, 4.724 Hz)

Glutamine

2.13 m; 2.45 m; 3.77 t (J=6.18 Hz)

Figure 3 and 4: showing volcano plots and heatmaps using features is meaningless without metabolite ID. The authors should integrate well-isolated metabolite signals and use these integrals for further analysis or determine concentrations/integrals using line shape fitting with programs such as Chenomx or other comparable software. 

Answer:  Figures 3 and 4 were substituted with the ones where the most important variables (chemical shifts) were identified and we have added their IDs in the revised  Figures’ versions. Unfortunately, we do not have a Chenomx subscription, but we have used MestreNova and TopSpin to integrate and process all spectra and compare the differences in concentrations in top metabolites among the studied samples.